# Beverage Intake and the Risk of Gestational Diabetes Mellitus: The SECOST

**DOI:** 10.3390/nu13072208

**Published:** 2021-06-27

**Authors:** Heng Yaw Yong, Zalilah Mohd Shariff, Barakatun Nisak Mohd Yusof, Zulida Rejali, Yvonne Yee Siang Tee, Jacques Bindels, Eline M. van der Beek

**Affiliations:** 1Department of Nutrition, Faculty of Medicine and Health Sciences, Universiti Putra Malaysia, Seri Kembangan 43400, Selangor, Malaysia; yong_hy@upm.edu.my; 2Department of Dietetics, Faculty of Medicine and Health Sciences, Universiti Putra Malaysia, Seri Kembangan 43400, Selangor, Malaysia; bnisak@upm.edu.my; 3Department of Obstetrics and Gynaecology, Faculty of Medicine and Health Sciences, Universiti Putra Malaysia, Seri Kembangan 43400, Selangor, Malaysia; zulida@upm.edu.my; 4Danone Specialized Nutrition (Malaysia) Sdn. Bhd., Mid Valley City, Lingkaran Syed Putra, Kuala Lumpur 59200, Malaysia; yvonneyeesiang.tee@danone.com; 5Nutricia Research Foundation, Conradpark 3, 2441 AE Nieuwvee, The Netherlands; jacquesbindels@yahoo.de; 6Department of Pediatrics, University Medical Centre Groningen, University of Groningen, Hanzeplein 1, 9713 GZ Groningen, The Netherlands; e.m.van.der.beek@umcg.nl

**Keywords:** cultured-milk drinks, fruit juice, Seremban Cohort Study, pre-pregnancy, first trimester

## Abstract

The contribution and impact of beverage intake to total nutrient and energy intake may be substantial. Given the link between lifestyle, diet, and the risk of pregnancy complications, this study investigated the association between the quantity and types of beverages with gestational diabetes mellitus (GDM) risk. The study included 452 women from the Seremban Cohort Study (SECOST). The mean energy by beverage intake was 273 ± 23.83 kcal/day (pre-pregnancy), 349 ± 69.46 kcal/day (first trimester) and 361 ± 64.24 kcal/day (second trimester). Women significantly increased intake of maternal milks and malted drinks, but significantly reduced the intake of carbonated drinks and other drinks from before until the second trimester of pregnancy. For chocolate drinks, carbonated drinks, and soy milk, women increased intake from pre-conception to the first trimester, but reduced their intake from the first to the second trimester. While higher intake of cultured-milk drinks was associated with an increased risk of GDM, higher fruit juice intake was associated with a lower risk of GDM. However, these associations were only observed for intake prior to pregnancy and during the first trimester. Further research is needed to corroborate these findings and investigate the contributions of different beverages to overall diet quality as well as adverse health outcomes during pregnancy.

## 1. Introduction

The contribution of sugar-sweetened beverages (SSB) is starting to receive more attention because of the possible links with adverse metabolic outcomes. SSBs are defined as any beverage that contains added sugar (e.g., cane sugar, granulated sugar, brown sugar, honey, dextrose), caloric sweetener (e.g., high-fructose corn syrup, sucrose), or fruit juice concentrates (by manufacturers, establishments, or individuals), and typically contain more than 25 calories per 8 fluid ounces [1,2]. These include the full range of non-diet carbonated/soft drinks, energy, and sports drinks, sweetened coffee or tea, flavored fruit drinks, and electrolyte replacement drinks [3]. Globally, the SSB consumption has increased significantly over the last three decades. In the United States (US), the SSB consumption increased threefold, whereby from 3.9% of calories in the late 1970′s to 9.2% in 2001 [4].

Previous large-scale epidemiological studies have consistently found positive associations between SSB consumption and long-term weight gain and the risk of developing chronic diseases, such as metabolic syndrome, type 2 diabetes mellitus (T2DM), and coronary artery disease (CHD) [5,6,7]. SSB consumption is thought to contribute to weight gain due to the added sugar content, low satiety, and incomplete compensation for total energy at subsequent meals following liquid calories intake [8]. SSB also appears to play a crucial role in glycemic load (GL) in that large quantities of SSB consumption could contribute to high GL and, subsequently, contribute to inflammation, insulin resistance, and impaired beta-cell function [9], all of which may increase the risk of T2DM.

A recent study that examined SSB consumption of non-pregnant and pregnant women showed that one-fifth (21.9%) had at least one intake of SSB daily, and non-married women had a greater likelihood of daily SSB intake than married women [10]. Gamba et al. (2019) found that SSB consumption among pregnant women was associated with a poorer diet quality and a greater total calorie intake [11]. Several studies have related SSB intake with adverse pregnancy outcomes, such as pre-eclampsia, preterm delivery, poor child’s physical, and cognitive development [12,13,14,15,16]. The Nurses’ Health Study II on SSB consumption and GDM risk showed that higher pre-pregnancy sugar-sweetened cola intakes (≥5 servings/week) were associated with increased GDM risk [17].

The Malaysian Adults Nutrition Survey found that 98.6% of Malaysian adults reported daily consumption of SSB, with 2 glasses per day on average. Tea (70.3%), malted drinks (59.1%), coffee (53.2%), soy milk (51.4%), and carbonated drinks (45.6%) were the top five consumed beverages [18]. Malaysians habitually consume beverages, such as tea, coffee, and malted drinks with added sugar, and/or sweetened creamer (non-dairy / dairy) [19], which may lead to additional energy consumption. Limited evidence suggests that energy-containing beverages increase the overall energy intake and subsequently the risk of chronic diseases. However, little is known regarding the intake of beverages during pregnancy or its influence on pregnancy complications, such as GDM. The aim of this study was to determine the types and quantity of beverages consumed by pregnant women in Malaysia, as well as to investigate the possible associations between beverage consumption with GDM risk.

## 2. Materials and Methods

### 2.1. Study Design and Respondents

Respondents were pregnant women from three maternal child health (MCH) clinics in Seremban district, Negeri Sembilan, who were enrolled in the Seremban Cohort Study (SECOST), a multisite prospective cohort study aimed to identify the determinants and pregnancy outcomes of maternal glycemia. Details of the SECOST study have been described previously and the present study reported on data of 452 pregnant women [20]. All women provided informed consent prior to study enrollment.

### 2.2. Measurements

#### 2.2.1. Blood Glucose and Diagnosis of Gestational Diabetes Mellitus

GDM diagnosis, outlined in the Perinatal Care Manual Third Edition [21], was based on a standard two-point diagnostic 75 g oral glucose tolerance test (OGTT) performed between 28 and 32 weeks of gestation. Results were interpreted as normal glycemia (both FPG < 5.6 mmol/L and 2 hPG < 7.8 mmol/L) or diagnostic for GDM (either or both FPG was ≥5.6 mmol/Lor 2 hPG ≥ 7.8 mmol/L).

#### 2.2.2. Beverage Intake

Women were requested to complete a validated 126-food item semi-quantitative Food Frequency Questionnaire (SFFQ) at the first prenatal visit (9.82 ± 2.51 gestational weeks) for food intake before pregnancy, to the first trimester (12.26 ± 1.58 gestational weeks) and the second trimester (26.73 ± 1.64 gestational weeks), respectively. Nutritionist Pro Diet Analysis software: Version 1.5 [22] with United States Department of Agriculture food database [23] was used to analyze dietary data. Total energy intake (kcal/day), total energy from beverages (kcal/day) and percentage of energy from beverages as well as total sugar intake (g/day), percentage of energy from total sugar intake (%), sugar intake from beverages (g/day), percentage of sugar intake from beverages (%), and percentage of energy from sugar derived from beverages (%) were calculated.

Intake of beverages (g/day) was estimated by multiplying the reported portion size consumed by the reported frequency of consumption. Beverages were aggregated into 11 groups: milks, tea, coffee, chocolate drinks, malted drinks, syrup/cordial, fruit juice (homemade or commercial), cultured-milk milks, carbonated drinks, soy milk, and other drinks [24]. Due to religious prohibition of alcohol consumption among Muslim, the overall consumption of alcohol in this sample of women (~90% Muslim) was very low. Therefore, alcoholic drinks were included into the ‘Other drinks’.

#### 2.2.3. Socio-Demographic and Obstetrical Information

Socio-demographic variables include age, education level, ethnicity, occupation status, and monthly household income. Medical records were used to obtain obstetrical information (e.g., parity, GDM medical history, and family history of diabetes mellitus).

#### 2.2.4. Anthropometric Measurements

A SECA digital weighing scale and SECA body meter were used to measure weight at each study visit and height at study enrolment, respectively. Women were asked to report their weight before pregnancy. Pre-pregnancy body mass index (BMI) (kg/m^2^) was determined as weight before pregnancy in kilograms divided by height in meters squared and further categorized based on the recommendation of the World Health Organization [25]. Total gestational weight gain (GWG) was computed by subtracting weight before pregnancy from weight at last prenatal visit and categorized according to the Institute of Medicine (IOM) guidelines [26].

### 2.3. Statistical Analysis

Statistical Package of Social Sciences (SPSS) version 26 was used to perform all analyses. Exploratory Data Analysis (EDA) was employed to test for data normality and homogeneity of variance. No transformation was performed as all continuous variables were normally distributed. Basic descriptive statistics such as means, standard deviations, frequency, and percentage distribution were used to describe the data. Logistic regression was performed to assess the association between the beverages intake (independent variables) and GDM risk (dependent variable) at each time point, adjusted for covariates that have the potential to confound the association between the beverage intakes and GDM risk. The included covariates were age (continuous), total energy intake (continuous), pre-pregnancy BMI (continuous), and total GWG (continuous). Non-GDM served as reference group. Adjusted odds ratios (AORs) were used to estimate the strength of the associations while 95% confidence interval (CI) were used for significance testing. Statistical significance was set at *p*-value < 0.05.

## 3. Results

### 3.1. Women’s Characteristics

Table 1 summarizes the characteristics of the women in this study. Most of the women were Malay (88.9%), 30 years old or below (60.4%), employed during the study period (68.6%), and had secondary or lower education (46.0%), and low household incomes (62.8% reported less than RM 3860 per month). One-third of the women were primigravida (35.4%), and another one-third had three or more pregnancies. About 6.9% of the women had a history of GDM, and 24.6% had a family history of DM. More than 30% of the women were either overweight (22.8%) or obese (11.3%). The mean total GWG was 11.40 ± 5.91 kg, with 38.7% showing inadequate weight gain and 23.0% gaining excessive weight. The mean total energy intake were 2159 ± 980.17 kcal/day for pre-pregnancy, 2033 ± 913.87 kcal/day for first trimester, and 2172 ± 896.90 kcal/day for second trimester. Meanwhile, the mean contribution to the total intake by beverages were 273 ± 3.83 kcal/day for pre-pregnancy, 349 ± 69.46 kcal/day for first trimester and 361 ± 64.24 kcal/day for second trimester. GDM was diagnosed in 10.6% (*n* = 48) of the women.

### 3.2. Beverage Intakes before and during Pregnancy

Beverage intakes before and during pregnancy are shown in Table 2. Women significantly increased the intake of maternal milks (*p* = 0.001), and malted drinks (*p* = 0.01), but significantly reduced the intake of carbonated drinks (*p* = 0.001) and other drinks (*p* = 0.03) from pre-pregnancy to the second trimester. For chocolate drinks (*p* = 0.04) and soy milk (*p* = 0.01), women increased their intakes from pre-conception to the first trimester, but reduced their intakes from the first to the second trimester. No significant associations were observed for other types of beverages, such as fresh milk/UHT, powdered milk (all type), tea, coffee, syrup/cordial, fruit juice (homemade or commercial), and cultured-milk drinks.

### 3.3. Associations between Beverage Intake and GDM Risk

Only fruit juice (homemade or commercial) and cultured-milk drinks showed modest, yet significant associations with GDM risk (Table 3). Women with higher fruit juice intake before pregnancy (AOR = 0.98, 95% CI = 0.97–0.99) and in the first trimester (AOR = 0.92, 95% CI = 0.89–0.98) were at slightly lower risk to develop GDM. Conversely, women with higher intake of cultured-milk drinks before pregnancy (AOR = 1.03, 95% CI = 1.01–1.08) and in the first trimester (AOR = 1.07, 95% CI = 1.02–1.12) were at significantly higher GDM risk. Appendix A shows the energy and macronutrient intakes of women by cultured-milk drinks and fruit juice categories. Women with the highest tertile of cultured-milk drinks intake before pregnancy had significantly higher intake of fat compared with women with the highest tertile of fruit juice intake. Although there were no significant associations between both groups for energy and other macronutrients in both pre-pregnancy and in the first trimester, women with the highest tertile of cultured-milk drinks intake had higher energy, fat and protein intakes before pregnancy and higher energy, and sugar intakes in the first trimester than women with the highest tertile of fruit juice intake. Women in both groups increased their sugar intakes in the first trimester but women with the highest tertile of cultured-milk drinks intake had a higher sugar intake than women with the highest tertile of fruit juice intake. Women with the highest tertile of cultured-milk drinks intake had diets characterized by more unhealthy food groups before pregnancy (e.g., tea, coffee, spreads, sweet foods, sugar, and creamer) and in the first trimester (e.g., processed foods, sauces, sugar, and creamer) (Appendix A). Women with the highest tertile of energy intake derived from beverages had significantly higher mean daily energy and macronutrient intakes than women with the lowest tertile (Appendix A).

## 4. Discussion

The role of sugar in diabetes and GDM development has generated much debate, and increasing epidemiologic evidence suggests that especially beverages containing fructose, including natural fruit juices, are associated with greater risk of type 2 diabetes (T2DM) [2,27,28] and GDM [29]. However, this study showed that women with higher fruit juice intake, either homemade or commercial, before pregnancy and in the first trimester were significantly at lower GDM risk. Notably, the protective association also persisted in the second trimester of pregnancy, albeit no longer significant. The association between fermented milk consumption and reduced T2DM risk has been consistently reported [30], but there are minimal investigations on the impacts of fermented milk consumption on GDM risk [31,32]. The present study found that women with higher intake of cultured-milk drinks before pregnancy and in the first trimester had significantly higher GDM risk, even after adjusting for covariates.

Further sub-group analysis of women with intakes of fruit juice, either homemade or commercial, and cultured-milk drinks, showed that women in the highest tertile of cultured-milk drinks intake before pregnancy had a less healthy dietary intake compared with women in the highest tertile of fruit juice intake. Thus, it is possible that the overall diet, a combination of cultured-milk drinks and unhealthy foods, could contribute to GDM risk. Conversely, some other components in food juices, such as vitamins, minerals, and phytochemicals at low to moderate level of consumption may compensate the adverse effects of the rapidly absorbed sugars. The observed association should, however, be interpreted with caution, as this study did not assess the type of fruits in the drinks (e.g., high sugary fruits vs. low sugary fruits), or the fiber content of fruits, as well as the types of fruit juices (e.g., fresh homemade vs. commercial fruit juices). The homemade juicing process might not remove the edible skin and pulp, which are sources of fiber and an essential nutrient that helps delay the absorption of sugar. Based on the available 24-h diet recall data for the women in this study, they were more likely to report consumption of homemade fruit juices (with no added sugar) rather than consumption of commercial fruit juices. The present study seems to support that low to moderate consumption of homemade fruit juices could be considered a healthy beverage because of the vitamins, minerals, and phytochemicals in fruit juices. Meanwhile, the effect of cultured-milk drinks on GDM risk requires further investigation as whether the association could be confounded not only by other nutrients in the overall diet, but also specific components in the cultured-milk drinks, such as sugar and fat content.

To date, limited information is available on the contribution of beverages to pregnant women’s energy intake. The present study found that the mean energy contributed by beverages were 273 ± 23.83 kcal/day (12.97 ± 9.70%) for pre-pregnancy, 349 ± 69.46 kcal/day (17.00 ± 10.29%) for the first trimester and 361 ± 64.24 kcal/day (16.60 ± 8.70%) for the second trimester, indicating that women increased their beverage calories during pregnancy. In the National Health and Nutrition Examination Survey (NHANES), Gamba et al. (2019) showed that pregnant women (176 kcal/day) had a higher energy contributed by SSBs than non-pregnant women (138 kcal/day) [11]. A lower energy contributed by SSBs was reported in this study with the mean energy contributed by SSBs was 104.13 ± 76.24 kcal/day for pre-pregnancy, 107.23 ± 89.02 kcal/day for the first trimester and 110.33 ± 86.92 kcal/day for the second trimester. However, caution should be exercised when comparing between studies as different beverage items are included in SSBs. Additional analysis of the present study data showed that women with the highest tertile of energy intake derived from beverages had significantly higher mean daily energy and macronutrient intakes than women with the lowest tertile. Although it is still unclear whether beverage intake suppresses the intake of other foods or increases the overall energy intake by serving as “add-on” energy, beverages could contribute substantially to total energy intake.

In the Nurses’ Health Study II, higher carbonated beverages consumption, especially sugar-sweetened cola (≥5 servings per week) was related to an elevated risk of GDM [17]. However, the present study did not support this association. Majority (50–70%) of women in this study reported no intake of carbonated drinks as pregnancy progressed from first to second trimester. For women reporting intake of carbonated drinks, the mean intake was <1 serving per week (1 serving equates to 8 fluid ounces or 236.59 mL). Another explanation is that this study assessed the overall intake of carbonated drinks, which did not distinguish the type of carbonated drinks (e.g., soda vs. diet soda, classic cola vs. light cola).

Caffeine consumption during pregnancy has been associated with adverse outcomes, such as impaired fetal length growth, low birth weight, and increased risk of delivering a small-for-gestational-age (SGA) infant [33,34,35,36]. In this study, women reduced their intakes of tea, coffee, and chocolate drinks from pre-pregnancy to the second trimester of pregnancy, although the decrease was not statistically significant. This finding was consistent with previous studies [37,38], whereby from pre-pregnancy to pregnancy, women significantly decreased their intakes of caffeine, particularly coffee. In non-pregnant adults, acute ingestion of caffeine induces insulin resistance, while habitual caffeine intake decreases sub-clinical inflammation and increases adiponectin levels, which may protect against insulin resistance and lower T2DM risk [39,40]. Studies also showed that antioxidant and prebiotic-like properties, such as chlorogenic, ferulic, caffeic, and *n*-coumaric acids found in coffee or tea tend to improve glucose control, insulin sensitivity, and appetite regulation [41]. The present study supports previous findings also showing non-significant associations between coffee or tea consumption in early pregnancy and GDM risk [42,43]. The low intake of these beverages among women in the present study, whereby most of the women consumed less than ½ cup per day of coffee or tea during pregnancy, may explain the absence of any significant association, despite its potential contribution to total sugar intake. According to the guidelines of the American College of Obstetrics and Gynecology (ACOG), women can consume one cup of coffee per day without an increased risk of miscarriage or preterm delivery. Thus, consumption of small amounts of coffee or tea (<1 cup) during pregnancy does not seem to have any negative impact on pregnancy outcomes.

Malted drinks are malted-based food products, a mixture of malt with other cereal and legume flour with or without whole milk or milk powder and/or cocoa powder [44]. The drinks are marketed as nutritious beverages due to their high nutrient contents (e.g., carbohydrate, protein, fats, vitamin A, B, C, and E, calcium, iron, phosphorus and potassium) [23,44,45]. Nevertheless, published literature on malted drinks in relation to disease outcomes is very limited [46], and none of the evidence relates to pregnant women. Although the present study did not find any significant association between malted drinks and the risk of GDM, the intake of malted drinks significantly increased from pre-pregnancy to the second trimester. Since it is one of the most popular beverages among Asians, it is worth investigating its relative contribution to maternal health in more detail.

This study has several limitations. The instrument could be subjected to recall and social desirability biases as this study used self-reported SFFQ. The details of beverages such as the types of fruits, homemade or commercial fruit juices and types of cultured-milk/carbonated drinks were not assessed. The study also did not allow estimating the actual amounts of sugar and sweetened creamer added into the tea or coffee, but amounts were estimated based on the common consumption pattern of Malaysian adults [47]. The accuracy of energy and sugar intake estimates might therefore be limited as the estimations were based on the USDA food database and food labels of fortified foods in the market. Although significant associations were observed between intakes of fruit juice and cultured-milk drinks intake with GDM risk, the effect size is relatively small. More research is needed to confirm these findings. Moreover, the present study did not obtain several important covariates, including glycemic index and glycemic load of beverages, which might contribute to the overall effects. As the study population was mainly composed of Malays, less educated, from low-income households that were employed during the study period, the study findings cannot be generalized to all Malaysian pregnant women.

## 5. Conclusions

While the consumption of cultured-milk drinks in early pregnancy showed a modest positive association with odds of developing GDM, intake of fruit juices in early pregnancy albeit in low quantity, either homemade or commercial, was modestly associated with odds of developing GDM. While the effect of cultured-milk drinks on GDM risk remains to be further explored, it is also unclear whether the protective association of fruit juice holds true for all types of fruits, and homemade and/or commercial fruit juices. Although fruit juices contain some valuable (micro) nutrients, and the study suggested that fruit juices seem to have no harmful effect on GDM risk, the consumption should be in moderate amounts (less than two servings per day). It is prudent that the dietary guidance on beverage intake during pregnancy considers the assessment of overall dietary intake.

## Figures and Tables

**Table 1 nutrients-13-02208-t001:** Sociodemographic, obstetrical, anthropometric, diet, and blood glucose information of women (*n* = 452).

Characteristics	*n* (%)	Mean ± SD
Ethnicity		
Malay	402 (88.9)	
Non-Malay	50 (11.1)	
Age at study entry (years)		30.01 ± 4.48
≤30	273 (60.4)	
>30	179 (39.6)	
Education (years)		12.96 ± 2.39
Secondary and lower	208 (46.0)	
STPM/matric/diploma/certificate	148 (32.7)	
Tertiary and above	96 (21.3)	
Occupation		
Housewife	142 (31.4)	
Working	310 (68.6)	
Household income (RM) ^1^		3726.74 ± 2050.96
Low (<3860)	284 (62.8)	
Middle (3860–8319)	154 (34.1)	
High (≥8320)	14 (3.1)	
Parity		1.22 ± 0.45
0	160 (35.4)	
1–2	135 (29.9)	
≥3	157 (34.7)	
History of GDM		
No	421 (93.1)	
Yes	31 (6.9)	
Family history of DM		
No	341 (75.4)	
Yes	111 (24.6)	
Height (m)		1.56 ± 0.06
Pre-pregnancy weight (kg)		58.14 ± 12.92
Pre-pregnancy BMI (kg/m^2^)		23.73 ± 4.80
Underweight (<18.5)	48 (10.6)	
Normal (18.5–24.9)	250 (55.3)	
Overweight (25.0–29.9)	103 (22.8)	
Obese (≥30.0)	51 (11.3)	
Total GWG (kg) by ^2^		11.40 ± 5.91
Underweight (<18.5)		13.64 ± 4.31
Normal (18.5–24.9)		12.33 ± 5.72
Overweight (25.0–29.9)		9.36 ± 5.98
Obese (≥30.0)		8.86 ± 6.09
Energy intake (kcal/day)		
Pre-pregnancy		2159 ± 980.17
First trimester		2033 ± 913.87
Second trimester		2172 ± 896.90
Energy from beverages (kcal/day)		
Pre-pregnancy		273 ± 23.83
First trimester		349 ± 69.46
Second trimester		361 ± 64.24
Percentage of energy from beverages (%)		
Pre-pregnancy		12.97 ± 9.70
First trimester		17.00 ± 10.29
Second trimester		16.60 ± 8.70
Total sugar intake (g/day)		
Pre-pregnancy		98.00 ± 5.69
First trimester		98.12 ± 5.71
Second trimester		94.83 ± 2.62
Percentage of energy from total sugar intake (%)		
Pre-pregnancy		15.80 ± 7.13
First trimester		17.39 ± 6.79
Second trimester		16.01 ± 5.66
Sugar intake from beverages (g/day)		
Pre-pregnancy		28.05 ± 9.75
First trimester		36.93 ± 8.02
Second trimester		39.90 ± 9.05
Percentage of sugar intake from beverages (%)		
Pre-pregnancy		33.45 ± 9.73
First trimester		37.84 ± 8.18
Second trimester		40.97 ± 7.29
Percentage of energy from sugar derived from beverages (%)		
Pre-pregnancy		8.78 ± 4.89
First trimester		7.45 ± 4.47
Second trimester		7.30 ± 4.05
Maternal glucose level		
Oral glucose tolerance test (OGTT)		
Gestational weeks at OGTT performed		28.00 ± 0.24
Fasting plasma glucose (FPG) (mmol/L)		4.36 ± 0.43
2-h plasma glucose (2 hPG) (mmol/L)		5.88 ± 1.42
GDM according to MOH criteria ^3^	48 (10.6)	

^1^ 1 USD = RM 4.18 during study period, ^2^ IOM gestational weight gain guidelines, ^3^ GDM according to MOH criteria, either of both FPG ≥ 5.6 mmol/Lor 2 hPG ≥ 7.8 mmol/L.

**Table 2 nutrients-13-02208-t002:** Types of beverages consumed before and during pregnancy.

Types of Beverages ^1,2^	Pre-Pregnancy	First Trimester (Weeks 10–13)	Second Trimester (Weeks 24–30)	*p*-Value
	Mean ± SE (g/day)			
Milk				
Fresh milk/UHT/Flavored milk	71.27 ± 5.90	84.40 ± 7.36	85.95 ± 7.08	0.10
Powdered milk (all types)	4.16 ± 0.51	3.95 ± 0.57	3.28 ± 0.49	0.20
Maternal milks	1.88 ± 0.51	10.03 ± 0.85	13.32 ± 0.84	0.001 **
Tea	99.41 ± 7.50	92.76 ± 7.72	88.45 ± 8.11	0.32
Coffee	43.04 ± 4.83	32.94 ± 4.28	33.45 ± 4.27	0.40
Chocolate drinks	34.38 ± 4.37	41.89 ± 5.16	26.96 ± 3.85	0.04 *
Malted drinks	103.16 ± 7.13	122.86 ± 7.00	132.04 ± 6.89	0.01 *
Syrup / cordial	43.01 ± 5.73	32.98 ± 3.33	40.78 ± 4.87	0.80
Fruit juice (homemade or commercial)	45.22 ± 3.11	47.38 ± 4.16	54.10 ± 5.68	0.09
Cultured-milk drinks	8.15 ± 1.05	7.14 ± 1.10	7.27 ± 1.18	0.08
Carbonated drinks	13.40 ± 1.60	6.21± 1.13	5.75 ± 1.20	0.001 **
Soy milk	31.28 ± 78.30	51.88 ± 4.74	48.03 ± 4.22	0.01 *
Other drinks ^3^	6.22 ± 1.48	5.42 ± 1.45	2.38 ± 0.73	0.03 *

^1^ 1 mL = 1 g; 1 cup = 200 mL. ^2^ Maternal milks: milk powdered exclusively for pregnant mothers and lactating mothers. Chocolate drinks: Cadbury (or other brands) chocolate drink, homemade chocolate drink. Malted drinks: Horlick, Milo, Ovaltine, Nestomalt. Cultured-milk drinks: cultured milks, yogurt drinks. ^3^ Other drinks: herbal drinks, energy drinks and alcohol drinks. Adjusted for total energy intake, * *p* < 0.05, ** *p* < 0.01. (The one-way ANCOVA (analysis of covariance)).

**Table 3 nutrients-13-02208-t003:** Associations between types and quantity of beverages and gestational diabetes mellitus (GDM).

Variables ^2^		GDM ^1^
		AOR [95% CI]
Fresh milk/UHT/flavored milk	Pre-pregnancy	1.01 [0.98–1.03]
	First trimester	1.01 [0.98–1.02]
	Second trimester	0.99 [0.96–1.01]
Powdered milk (all types)	Pre-pregnancy	1.02 [0.98–1.03]
	First trimester	0.99 [0.97–1.03]
	Second trimester	1.06 [0.98–1.09]
Maternal milks	Pre-pregnancy	1.01 [0.98–1.03]
	First trimester	0.94 [0.98–1.02]
	Second trimester	0.98 [0.98–1.02]
Tea	Pre-pregnancy	0.98 [0.94–1.01]
	First trimester	0.99 [0.96–1.01]
	Second trimester	0.99 [0.98–1.04]
Coffee	Pre-pregnancy	0.99 [0.98–1.01]
	First trimester	0.95 [0.98–1.01]
	Second trimester	0.99 [0.98–1.04]
Chocolate drinks	Pre-pregnancy	0.99 [0.95–1.03]
	First trimester	0.97 [0.93–1.02]
	Second trimester	0.99 [0.91–1.03]
Malted drinks	Pre-pregnancy	0.98 [0.95–1.01]
	First trimester	0.98 [0.96–1.01]
	Second trimester	1.02 [0.98–1.02]
Syrup/cordial	Pre-pregnancy	0.97 [0.92–1.03]
	First trimester	0.98 [0.99–1.03]
	Second trimester	1.12 [0.97–1.30]
Fruit juice (homemade or commercial)	Pre-pregnancy	0.98 [0.97–0.99] **
	First trimester	0.92 [0.89–0.98] *
	Second trimester	0.98 [0.93–1.02]
Cultured-milk drinks	Pre-pregnancy	1.03 [1.01–1.08] **
	First trimester	1.07 [1.02–1.12] **
	Second trimester	0.99 [0.96–1.02]
Carbonated drinks	Pre-pregnancy	0.99 [0.98–1.06]
	First trimester	0.98 [0.97–1.01]
	Second trimester	1.02 [0.96–1.03]
Soy milk	Pre-pregnancy	0.99 [0.95–1.04]
	First trimester	0.97 [0.93–1.02]
	Second trimester	0.76 [0.01–1.04]
Other drinks (herbal, energy, alcohol)	Pre-pregnancy	0.98 [0.94–1.02]
	First trimester	0.99 [0.98–1.03]
	Second trimester	0.98 [0.93–1.04]

^1^ Normal glycemia as reference group. ^2^ Adjusted by age, parity, total energy, pre-pregnancy BMI, and total GWG. * *p* < 0.05, ** *p* < 0.01.

## Data Availability

The data presented in this study are available on request from the corresponding author. The data are not publicly available as the Medical Research Ethics Committee (MREC), Ministry of Health Malaysia, imposed restrictions of disclosure data containing potentially identifying patient information.

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
