# Peer review of "Beverage Intake and the Risk of Gestational Diabetes Mellitus: The SECOST"

_nutrients, 2021, doi:10.3390/nu13072208_

Round 1
Reviewer 1 Report
Dear authors
The scientific paper “Beverage intake and the risk of gestational diabetes mellitus: The SECOST” although the low originality of the study, as there are already some studies with the same aim, it is important because it was carried out in Malaysian pregnant women population that has not yet been studied for this issue.
Studying subjects related to diet, which varies from person to person and from population to population, is hard and when it is not done rigorously, it can lead to results with many deviations.
The association between the ingestion of a certain food and a health risk is a very difficult issue, laborious and not always with promising results.
However, even with these limitations, this study gives indication and information to be taken into account and the results obtained could be the object of new and more refined studies.
Here are my suggestions / corrections
Abstract
Line 25 – the carbonated drinks reduced the intake as for other drinks from pre-pregnancy until the second trimester.
Keywords
Line 32 – keywords should be different to the words of the title.
- Materials and Methods
2.1 Study Design and Respondents
Lines 82 and 87 – repeated.
2.2.2 Beverage intake
Line 101 – No reference to the Total sugar intake (g/day), Percentage of energy from total sugar intake (%), Sugar intake from beverages (g/day), Percentage of sugar intake from beverages (%), Percentage of energy from sugar derived from beverages (%) calculations that appears on table 1.
Line 106 and 107 – The references 30 and 31 are not in order.
2.3. Statistical Analysis
Lines 140 and 141 - It would be important to explain how to interpret these results of the multivariable-adjusted odds ratio (OR) to be more understandable.
- Results
3.1. Characteristics of Women
Line 152 – WHO, 1998 - unnumbered reference.
Line 161 - the caption could be more complete.
Table 1
Education level - 12.96 ± 2.39 - How is this value calculated, what does it mean?
Total GWG (kg) - It would be interesting to put the interval here for each situation, as you put it in BMI.
3.2. Beverage Intakes Before and During Pregnancy
Line 170 - In the case of carbonated drinks there are a reduction from the pre-pregnancy to the first and the second trimester, as for other drinks.
3.3. Associations Between Beverage Intake and GDM Risk should be Associations between Beverage Intake and GDM Risk
Line 182 - Adjusted odds ratio (AOR), please add the acronym.
Lines 183 to 189 – Hope the explanation of the OR in lines 140 and 141 helps to understand better these phrases. “Only fruit juice (homemade or commercial) and cultured-milk drinks showed modest, yet significant associations with the risk of GDM. Women with higher fruit juice intake before pregnancy (AOR= 0.98, 95% CI= 0.97 – 0.99) and in the first trimester (AOR= 0.92, 95% CI= 0.91 – 0.98) were at slightly lower risk to develop GDM. Conversely, women with higher intake of cultured-milk drinks before pregnancy (AOR= 1.03, 187 95% CI= 1.01 – 1.08) and in the first trimester (AOR= 1.07, 95% CI= 1.02 – 1.12) were at significantly higher risk of GDM”. The modest and higher association is because the p value is 0.05 and 0.01 and not because the values of the AOR.
Table 3 – Hope that explanation of the OR in lines 140 and 141 helps in better interpretation of these table 3 results.
- Discussion
With regard to the discussion of the results regarding the supplementary material, I was unable to carry out an evaluation, as I was unable to access this material.
Author Response
RESPONSE TO REVIEWERS
Reviewer 1
Dear authors
The scientific paper “Beverage intake and the risk of gestational diabetes mellitus: The SECOST” although the low originality of the study, as there are already some studies with the same aim, it is important because it was carried out in Malaysian pregnant women population that has not yet been studied for this issue. Studying subjects related to diet, which varies from person to person and from population to population, is hard and when it is not done rigorously, it can lead to results with many deviations. The association between the ingestion of a certain food and a health risk is a very difficult issue, laborious and not always with promising results. However, even with these limitations, this study gives indication and information to be taken into account and the results obtained could be the object of new and more refined studies.
Response:
Thank you for the positive comment.
- Abstract
Line 25 – the carbonated drinks reduced the intake as for other drinks from pre-pregnancy until the second trimester.
Response:
We have revised accordingly. (line 24-25, text in red colour)
"Women significantly increased intake of maternal milks and malted drinks, but significantly reduced the intake of carbonated drinks and other drinks from before until the second trimester of pregnancy."
- Keywords
Line 32 – keywords should be different to the words of the title.
Response:
We have revised accordingly. (line 33, text in red colour)
Keywords: cultured-milk drinks; fruit juice; Seremban Cohort Study; pre-pregnancy, first trimester
- Materials and Methods
2.1 Study Design and Respondents
Lines 82 and 87 – repeated.
Response:
The repeated sentence has been deleted.
- 2.2 Beverage intake
Line 101 – No reference to the Total sugar intake (g/day), Percentage of energy from total sugar intake (%), Sugar intake from beverages (g/day), Percentage of sugar intake from beverages (%), Percentage of energy from sugar derived from beverages (%) calculations that appears on table 1.
Response:
We have revised accordingly. (line 101-105, text in red color)
Total energy intake (kcal/day), total energy from beverages (kcal/day) and percentage of energy from beverages as well as total sugar intake (g/day), percentage of energy from total sugar intake (%), sugar intake from beverages (g/day), percentage of sugar intake from beverages (%), and percentage of energy from sugar derived from beverages (%) were calculated.
- Line 106 and 107 – The references 30 and 31 are not in order.
Response:
We have revised the references accordingly. (line 100-102, reference 22 & 23)
- 3. Statistical Analysis
Lines 140 and 141 - It would be important to explain how to interpret these results of the multivariable-adjusted odds ratio (OR) to be more understandable.
Response:
We have revised accordingly. (line138-141, text in red colour)
"Non-GDM served as reference group. Adjusted odds ratio (AOR)s were used to estimate the strength of the associations while 95% confidence interval (CI) were used for significance testing. Statistical significance was set at p-value < 0.05."
- Results
3.1. Characteristics of Women
Line 152 – WHO, 1998 - unnumbered reference.
Response:
We have revised the sentence. (line 125, reference 25)
- Line 161 - the caption could be more complete.
Response:
We have revised the reference accordingly. (line 157-158, text in red colour)
Table 1. Socio-demographic, obstetrical, anthropometric, diet and blood glucose information of women (n=452)
- Table 1
Education level - 12.96 ± 2.39 - How is this value calculated, what does it mean?
Response:
We have revised the variable to “Education (years)”. (Table 1)
- Total GWG (kg) - It would be interesting to put the interval here for each situation, as you put it in BMI.
Response:
We have included the total GWG (kg) by pre-pregnancy BMI categories. (Table 1)
- 2. Beverage Intakes Before and During Pregnancy
Line 170 - In the case of carbonated drinks there are a reduction from the pre-pregnancy to the first and the second trimester, as for other drinks.
Response:
We have revised accordingly. (line 168, text in red colour)
Women significantly increased the intake of maternal milks (p= 0.001), and malted drinks (p= 0.01), but significantly reduced the intake of carbonated drinks (p= 0.001) and other drinks (p= 0.03) from pre-pregnancy to the second trimester.
- 3. Associations Between Beverage Intake and GDM Risk should be Associations between Beverage Intake and GDM Risk
Response:
We have revised accordingly. (line 180, text in red colour)
Line 182 - Adjusted odds ratio (AOR), please add the acronym.
Response:
We have added adjusted odds ratio (AOR) in the Statistical Analysis section. (line 139, text in red colour)
- Lines 183 to 189 – Hope the explanation of the OR in lines 140 and 141 helps to understand better these phrases. “Only fruit juice (homemade or commercial) and cultured-milk drinks showed modest, yet significant associations with the risk of GDM. Women with higher fruit juice intake before pregnancy (AOR= 0.98, 95% CI= 0.97 – 0.99) and in the first trimester (AOR= 0.92, 95% CI= 0.91 – 0.98) were at slightly lower risk to develop GDM. Conversely, women with higher intake of cultured-milk drinks before pregnancy (AOR= 1.03, 187 95% CI= 1.01 – 1.08) and in the first trimester (AOR= 1.07, 95% CI= 1.02 – 1.12) were at significantly higher risk of GDM”. The modest and higher association is because the p value is 0.05 and 0.01 and not because the values of the AOR. Table 3 – Hope that explanation of the OR in lines 140 and 141 helps in better interpretation of these table 3 results.
Response:
We have included explanations in Statistical analysis (refer to response in question 6).
- Discussion
With regard to the discussion of the results regarding the supplementary material, I was unable to carry out an evaluation, as I was unable to access this material.
Response:
As suggested by reviewer 2, we have included the findings of the supplementary tables in the Results section. (line 187-203, text in red colour)
Reviewer 2 Report
Suggest re-labeling “p-trend” to “p-value” in Table 2.
Line 215: Where is the data to support th following statement? “Women in both groups increased their sugar 215 intakes in the first trimester but women in the highest intake tertile of cultured-milk drinks had a higher sugar intake than women in the highest intake tertile of fruit juice.” If included in the supplementary data, this should still be described in the main manuscript under Results.
Given that almost ¼ of the participants gained excessive weight gain, the authors should consider adjusting for gestational weight gain for the analyses examining risk for GDM and beverage intake. It will be important to the field to know whether weight gain mediated any risk associations.
The authors should consider adding to the discussion their interpretation of the change in consumption of malted drinks increasing significantly over the course of pregnancy.
I would suggest the authors switch the paragraph starting at line 257 with the paragraph starting at 279.
Author Response
RESPONSE TO REVIEWERS
Reviewer 2
- Suggest re-labeling “p-trend” to “p-value” in Table 2.
Response:
We have revised accordingly. (Table 2)
- Line 215: Where is the data to support the following statement? “Women in both groups increased their sugar 215 intakes in the first trimester but women in the highest intake tertile of cultured-milk drinks had a higher sugar intake than women in the highest intake tertile of fruit juice.” If included in the supplementary data, this should still be described in the main manuscript under Results.
Response:
We have included the findings of the supplementary tables in the Results section (line 187-203, text in red colour) and revised the discussion (221-224, text in red colour) accordingly.
Results:
Only fruit juice (homemade or commercial) and cultured-milk drinks showed modest, yet significant associations with GDM risk (Table 3). Women with higher fruit juice intake before pregnancy (AOR= 0.98, 95% CI= 0.97 – 0.99) and in the first trimester (AOR= 0.92, 95% CI= 0.91 – 0.98) were at slightly lower risk to develop GDM. Conversely, women with higher intake of cultured-milk drinks before pregnancy (AOR= 1.03, 95% CI= 1.01 – 1.08) and in the first trimester (AOR= 1.07, 95% CI= 1.02 – 1.12) were at significantly higher GDM risk. Supplementary Table 1 shows the energy and macronu-trient intakes of women by cultured-milk drinks and fruit juice categories. Women with the highest tertile of cultured-milk drinks intake before pregnancy had significantly higher intake of fat compared with women with the highest tertile of fruit juice intake. Although there were no significant differences between both groups for energy and other macronutrients in both pre-pregnancy and in the first trimester, women with the highest tertile of cultured-milk drinks intake had higher energy, fat and protein intakes before pregnancy and higher energy and sugar intakes in the first trimester than women with the highest tertile of fruit juice intake. Women in both groups increased their sugar intakes in the first trimester but women with the highest tertile of cultured-milk drinks intake had a higher sugar intake than women with the highest tertile of fruit juice intake. Women with the highest tertile of cultured-milk drinks intake had diets characterized by more unhealthy food groups before pregnancy (e.g. tea, coffee, spreads, sweet foods, sugar and creamer) and in the first trimester (e.g. processed foods, sauces, sugar and creamer) (Supplementary Table 2). Women with the highest tertile of energy intake derived from beverages had significantly higher mean daily energy and macronutrient intakes than women with the lowest tertile (Supplementary Table 3).
Discussion:
Further sub-group analysis of women with intakes of fruit juice, either homemade or commercial, and cultured-milk drinks, showed that women in the highest tertile of cultured-milk drinks intake before pregnancy had a less healthy dietary intake compared with women in the highest tertile of fruit juice intake.
- Given that almost ¼ of the participants gained excessive weight gain, the authors should consider adjusting for gestational weight gain for the analyses examining risk for GDM and beverage intake. It will be important to the field to know whether weight gain mediated any risk associations.
Response:
We have revised Table 3, by including total GWG as covariate.
- The authors should consider adding to the discussion their interpretation of the change in consumption of malted drinks increasing significantly over the course of pregnancy.
Response:
We have included a paragraph regarding malted drinks as suggested. (line 290-299, text in red colour)
Malted drinks are malted-based food products, a mixture of malt with other cereal and legume flour with or without whole milk or milk powder and/or cocoa powder [44]. The drinks are marketed as nutritious beverages due to their high nutrient contents (e.g., carbohydrate, protein, fats, vitamin A, B, C and E, calcium, iron, phosphorus and potassium) [23,44,45]. Nevertheless, published literature on malted drinks in relation to disease outcomes is very limited [46], and none of the evidence relates to pregnant women. Although the present study did not find any significant association between malted drinks and the risk of GDM, the intake of malted drinks significantly increased from pre-pregnancy to the second trimester. Since it is one of the most popular beverages among Asians, it is worth investigating its relative contribution to maternal health in more detail.
- I would suggest the authors switch the paragraph starting at line 257 with the paragraph starting at 279.
Response:
We have switched the paragraph as suggested.